# Stage-specific expression patterns and co-targeting relationships among miRNAs in the developing mouse cerebral cortex
Hristo Todorov [1], Stephan Weißbach [1,2], Laura Schlichtholz[1,3], Hanna Mueller[1], Dewi Hartwich[1], Susanne Gerber [1] ✉ & Jennifer Winter[1] ✉

microRNAs are crucial regulators of brain development, however, miRNA regulatory networks are not sufficiently well characterized. By performing small RNA-seq of the mouse embryonic cortex at E14, E17, and P0 as well as in neural progenitor cells and neurons, here we detected clusters of miRNAs that were co-regulated at distinct developmental stages. miRNAs such as miR-92a/b acted as hubs during early, and miR-124 and miR-137 during late neurogenesis. Notably, validated targets of P0 hub miRNAs were enriched for downregulated genes related to stem cell proliferation, negative regulation of neuronal differentiation and RNA splicing, among others, suggesting that miRNAs are particularly important for modulating transcriptional programs of crucial factors that guide the switch to neuronal differentiation. As most genes contain binding sites for more than one miRNA, we furthermore constructed a co-targeting network where numerous miRNAs shared more targets than expected by chance. Using luciferase reporter assays, we demonstrated that simultaneous binding of miRNA pairs to neurodevelopmentally relevant genes exerted an enhanced transcriptional silencing effect compared to single miRNAs. Taken together, we provide a comprehensive resource of miRNA longitudinal expression changes during murine corticogenesis. Furthermore, we highlight several potential mechanisms through which miRNA regulatory networks can shape embryonic brain development.

Mammalian brain development is an extraordinarily complex process where neural progenitor cells proliferate and give rise to all neuronal and glial cell types. Numerous transcriptional, post-transcriptional, and epigenetic mechanisms integrate with each other to control progenitor proliferation and self-renewal, differentiation, and lineage commitment as well as migration in a strictly spatio-temporal manner. One of the key regulators of these processes are microRNAs (miRNAs). In most cases, these small RNAs bind to their target mRNAs' 3'- untranslated region (3'UTR) to induce mRNA degradation or translational inhibition. In the canonical pathway, miRNA genes are transcribed into a pri-miRNA in the cell nucleus by PolII/III[1,2]. In subsequent processing steps, the pri-miRNA is cleaved into a hairpin pre-miRNA by the microprocessor complex Drosha-Dgcr8 and transported to the cytoplasm by Exportin-5[3–6]. In the cytoplasm, the pre-miRNA is bound and cleaved by a complex containing the RNase Dicer1 to form the mature miRNA duplex[7]. From this duplex, the functional strand is loaded into the RISC complex,

guiding it to the 3'UTR's target site[8]. Mice carrying conditional *Dicer* gene deletions in the embryonic telencephalon have shown a variety of phenotypes in the cerebral cortex, such as reduced cell proliferation and impaired neuronal differentiation, increased apoptosis, defective cortical layering and microcephaly[9–14].

In the brain, 70% of all miRNAs are expressed, most of them in a highly cell-type- and developmental-time-specific manner[15]. Given that the total number of human miRNAs is estimated to be ~2300, at least 1600 different miRNAs may be expressed in the brain[16]. While this suggests that miRNA regulation is generally important for brain development and function, the specific functions of most miRNAs remain unknown. Single miRNAs are involved in some of these processes including, for example, cell proliferation (e.g., let-7, miR-124, miR-9), neuronal differentiation (e.g., let-7, miR-124, miR-9, miR-128) and migration (e.g., miR-9, miR-124, miR-379–410)[17,18]. Many of the functionally studied miRNAs are among the most highly expressed in the brain such as miR-9, miR-124 and let-7[17]. Most miRNAs

[1]Institute of Human Genetics, University Medical Center of the Johannes Gutenberg University Mainz, Mainz, Germany. [2]Institute of Developmental Biology and Neurobiology (iDN), Johannes Gutenberg University Mainz, Mainz, Germany. [3]Focus Program of Translational Neurosciences, University Medical Center Mainz, Mainz, Germany. ✉e-mail: sugerber@uni-mainz.de; jewinter@uni-mainz.de

typically have a relatively weak effect on their target genes[19,20]. Previous research has suggested that the regulatory potential of a given miRNA could be enhanced by cooperating with other miRNAs in co-targeting networks[21]. This hypothesis is supported by the fact that most 3'UTRs contain more than one miRNA binding site and most miRNAs have matching binding sites in up to several hundred 3'UTRs. A network of miRNAs co-operatively binding at distinct 3'UTR positions can potentially enhance target gene repression by additive or synergistic effects[22,23]. MiRNA binding at closely spaced binding sites (∼15–35 nt) can further enhance the repressive effect[23–25]. We and others have previously identified mRNAs that are targeted by several miRNAs leading to a synergistic or additive repressive effect[19,20,26]. For example, miRNAs of the miR-379-410 cluster regulate neurogenesis by targeting multiple miRNA binding sites in the N-cadherin 3'UTR in an additive manner[19]. While this highlights the regulatory potential of such miRNA networks during brain development, their detailed composition and temporal dynamics remain poorly characterized. Using an unbiased bioinformatics approach to identify co-targeting networks in mouse tissues, Cherone and colleagues recently found that miRNAs enriched in the human prefrontal cortex had more co-targeting partners than those enriched in other tissues suggesting that miRNA co-targeting is especially important in the brain[20].

In the current study, we employed high-throughput small RNA sequencing of the mouse cerebral cortex to create a detailed map of miRNA expression patterns and their longitudinal dynamics at key developmental stages. Using miRNAs whose expression was significantly correlated with developmental time, we constructed a comprehensive co-targeting network in which miRNAs with higher expression levels were associated with significantly more co-targeting relationships. In luciferase assays, we validated the enhanced gene silencing effect of cooperative miRNA binding in a set of target genes involved in the regulation of nervous system development.

## Results

### Temporal dynamics of miRNA expression during cerebral cortex development

To study the expression patterns of miRNAs during mammalian corticogenesis, we performed small bulk RNA sequencing in both female and male mice at a progenitor-dominated developmental phase (embryonic day E14), at an intermediate time point (E17) and at birth (postnatal day P0) when neurons from all six layers of the neocortex have been born[27,28] (Fig. 1). Using stringent filtering criteria for expression levels (>10 CPM in at least five samples), we detected a total of 489 miRNAs with only 13 of those corresponding to novel miRNAs. In agreement with previous reports[17,29], miR-9, members of the let-7 family, miR-128, and miR-124 were among the top 20 most abundantly expressed miRNAs in the developing cerebral cortex (Supplementary Fig. 1). Principal component analysis (PCA) revealed a distinct separation of samples according to their developmental stage. However, E17 and P0 samples had a more similar global expression pattern to each other compared to E14 (Fig. 2a). Since we did not observe apparent differences related to sex, female and male samples were processed together in the subsequent analyses.

We detected the highest number of differentially expressed miRNAs at the E14 vs. P0 time points, where 169 miRNAs were up- and 172 were down-regulated (Fig. 2b–f, Supplementary data 1). However, the major transcriptional shift already occurred in the transition from E14 to E17, where ∼57% of all detected miRNAs significantly changed their expression level (122 up- and 144 down-regulated miRNAs). In contrast, only 36% of all detected miRNAs were differentially expressed in E17 vs. P0 samples (82 upregulated and 93 downregulated miRNAs). Interestingly, a higher number of miRNAs were consistently downregulated at earlier developmental time-points in all pairwise comparisons (45 vs. 32 upregulated miRNAs, Fig. 2b, c, g).

To study the expression of miRNAs specifically during neurogenesis, we isolated NPCs from the cortices of mouse embryos at E14 and differentiated them in vitro into neurons. Small RNA sequencing of these cells

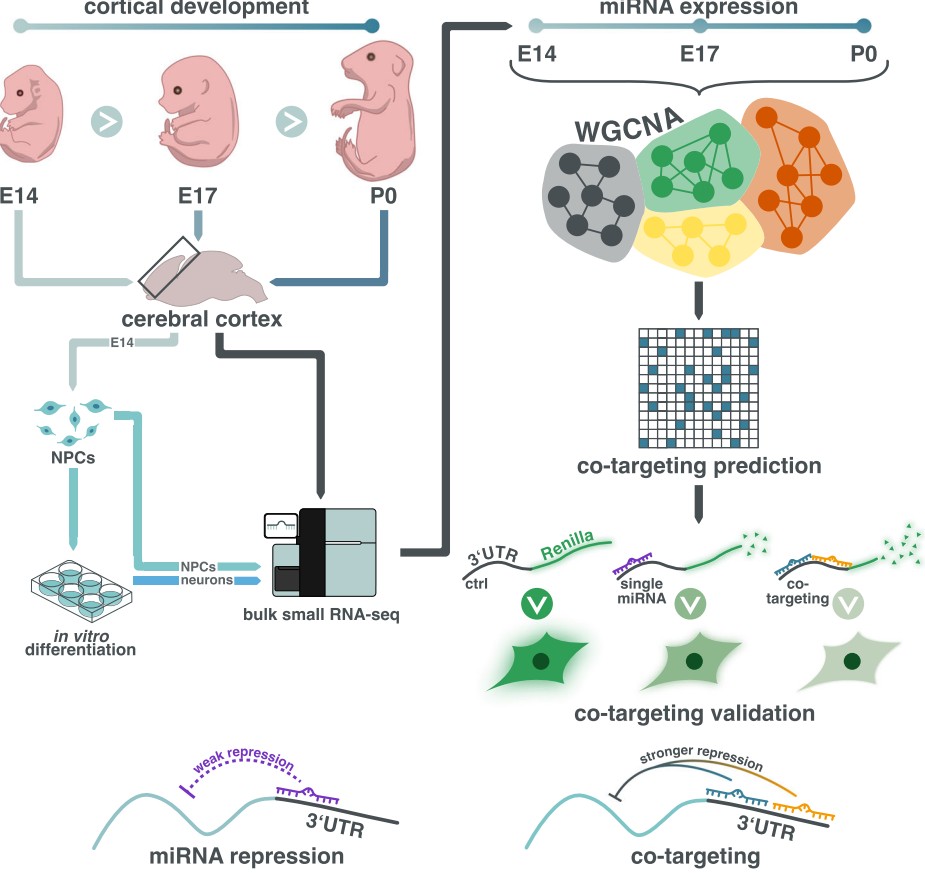

**Fig. 1 | Schematic representation of the study.** To investigate the expression patterns of miRNAs during cerebral cortical development in the mouse, we performed RNA-seq of bulk tissue at E14, E17 and P0 as well as in NPCs isolated from the cerebral cortex and differentiated into neurons. Subsequently, we employed weighted gene co-expression network analysis (WGCNA) to identify modules of co-expressed miRNAs. We then constructed a network of miRNAs sharing more target genes than expected by chance. Co-targeting relationships between selected miRNAs were validated using luciferase reporter assays, demonstrating enhanced gene silencing effect on neurodevelopmentally-relevant genes for combinations of miRNAs compared to individual miRNAs.

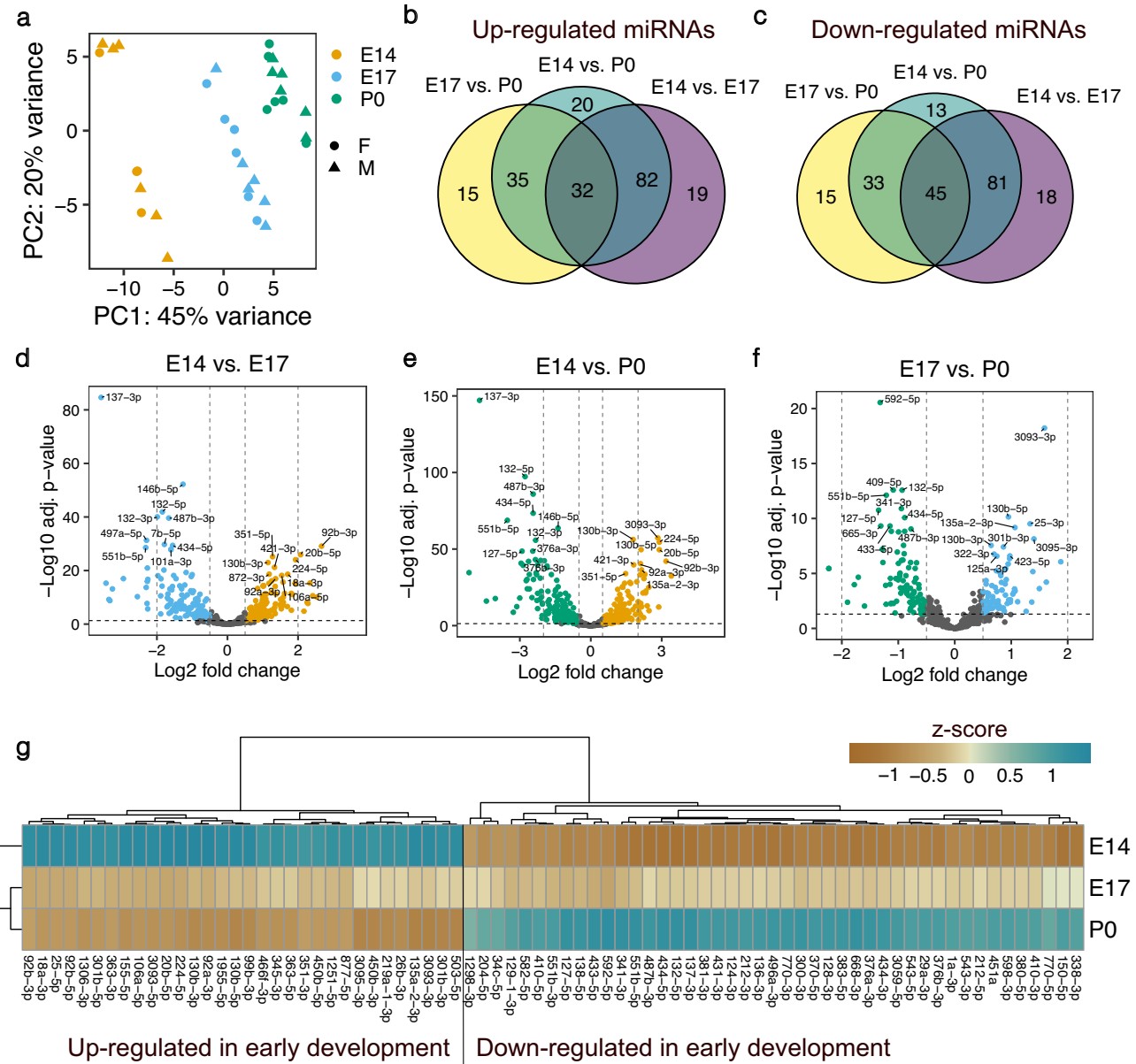

**Fig. 2 | Expression changes of miRNAs during mouse cerebral cortex development. a** Principal component analysis of bulk small RNA sequencing samples from the cerebral cortex of female and male mice at different developmental stages. $n = 4$ biological replicates for females at E14, $n = 6$ in all remaining groups. **b** Venn-diagram of upregulated miRNAs at earlier compared to later stages of cortical development. **c** Venn-diagram of downregulated miRNAs at earlier compared to later developmental stages. **d–f** Volcano plots of differentially expressed miRNAs. Positive log2 fold changes indicate miRNAs upregulated at earlier compared to later developmental stages in each pairwise comparison (E14 vs. E17, E14 vs. P0 and E17 vs. P0). **g** Heatmap of miRNAs differentially expressed in each pairwise comparison of different developmental stages. Mean expression values per developmental stage are shown as z-scores.

revealed 110 miRNAs that were upregulated and 113 miRNAs that were downregulated in NPCs compared to neurons, respectively. miR-124-3p and miR-369-3p were the most significantly upregulated miRNA in neurons, whereas miR-155-3p and miR-34b-3p were the most significantly upregulated miRNAs in NPCs (Supplementary Fig. 2a, d, Supplementary data 1). 66% of the upregulated (72 out of 110) and 65% of the downregulated (73 out of 113) miRNAs in NPCs vs. neurons were also differentially expressed in the comparison of bulk samples from E14 vs. P0, confirming the replicability of the expression patterns we observed (Supplementary Fig. 2b–d). Interestingly, these overlapping miRNAs corresponded to 43% of all upregulated and 42% of the downregulated miRNAs in E14 vs. P0, suggesting that a large proportion of the differentially regulated miRNAs might be attributable to the non-neuronal cells in the bulk samples. This observation is plausible, as gliogenesis takes place roughly between E17 and P0 of cortical development[28,30] therefore the bulk samples

from P0 likely contained both neuronal and non-neuronal cell types. In agreement, several of the developmental-stage-dependent expression changes we observed align with previously reported roles of miRNAs in inhibiting or promoting gliogenesis. For instance, miR-106a-5p was upregulated at E14 compared to E17 and P0 (Fig. 2g). miR-106a was previously reported to suppress gliogenic differentiation of neural stem cells and induce neurogenic cell fate commitment[31], which fits well with the upregulation of expression during peak neurogenesis in our study. Furthermore, we observed an upregulation of the gliogenic miR-338[32] specifically at P0 but not in neurons compared to NPCs (Supplementary data 1), thus underscoring the potential role of this miRNA specifically in promoting differentiation of glia cells. As another example, miR-153 whose inhibition was shown to confer gliogenic competence to neural stem cells[33], was upregulated in neurons compared to NPCs in our analysis (Supplementary data 1).

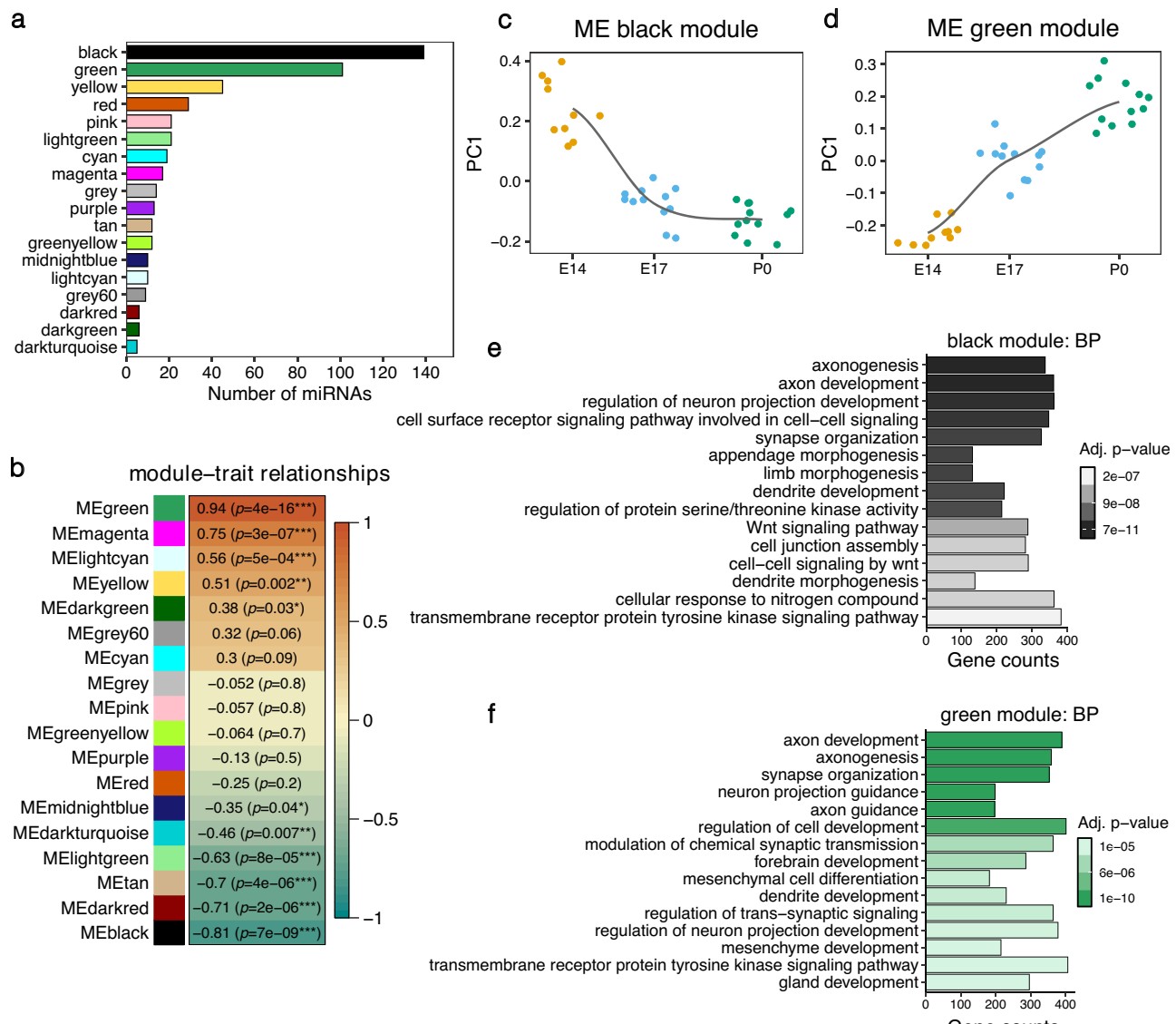

**Fig. 3 | Weighted miRNA co-expression analysis.** The co-expression network was derived using bulk small RNA-seq from mouse cerebral cortices at E14, E17 and P0. **a** Bar plot showing the number of miRNAs in each co-expression network module. **b** Pearson correlation between developmental stages of the mouse cerebral cortex (E14, E17 and P0) and the module eigengenes (ME). ***$p < 0.001$, **$p < 0.01$,

*$p < 0.05$. **c, d** Module eigengene values at each developmental stage for the black and green module. $n = 10$ biological replicates at E14, $n = 12$ in all remaining groups. **e, f** Top 15 enriched GO terms for the biological process (BP) category for the gene targets of the black and green modules.

## Weighted gene co-expression gene network analysis identifies sets of co-regulated miRNAs during the development of the cerebral cortex

Next, we applied weighted gene co-expression network analysis (WGCNA) to construct networks of co-expressed miRNAs and better characterize their transcriptional dynamics along the cortical developmental trajectory (Fig. 1). We obtained 18 modules that were assigned to arbitrary colors (Fig. 3a, detailed information of the specific miRNAs included in each WGCNA module is included in Supplementary data 2). Module eigengenes (MEs) that correspond to the first dimension in a principal component analysis of the expression matrix of the corresponding WGCNA module, served as proxies for the characteristic transcriptomic signature of each module. Six MEs were significantly negatively correlated with the developmental time point indicating that the miRNAs in the respective modules showed overall reduced expression levels at later compared to earlier developmental stages (Fig. 3b). In contrast, four modules contained miRNAs whose eigengene expression significantly increased during brain

development. The black module, which contained the highest number of miRNAs (139), had the strongest negative correlation with the developmental stage. The overall expression pattern captured by the ME, indicated that the black module contained miRNAs whose expression level peaked at E14 and dropped at E17, remaining stable afterwards (Fig. 3c). The green module was the second largest, containing 101 miRNAs with generally linearly increasing expression levels from E14 to P0 (Fig. 3d). Remarkably, although these two modules were regulated in opposite directions, they shared most of their targets (6244 common targets, corresponding to 79% of all black module and 72% of all green module targets, Supplementary Fig. 3c). Gene ontology analysis using the genes targeted by the miRNAs revealed that both the black and the green modules are involved in the regulation of key nervous system developmental processes such as axon development and guidance, neuron projection, dendrite development, and synapse organization (Fig. 3e, f, Supplementary data 3).

To identify key miRNAs at early and late embryonic brain developmental stages, we reconstructed the network structure of miRNAs within

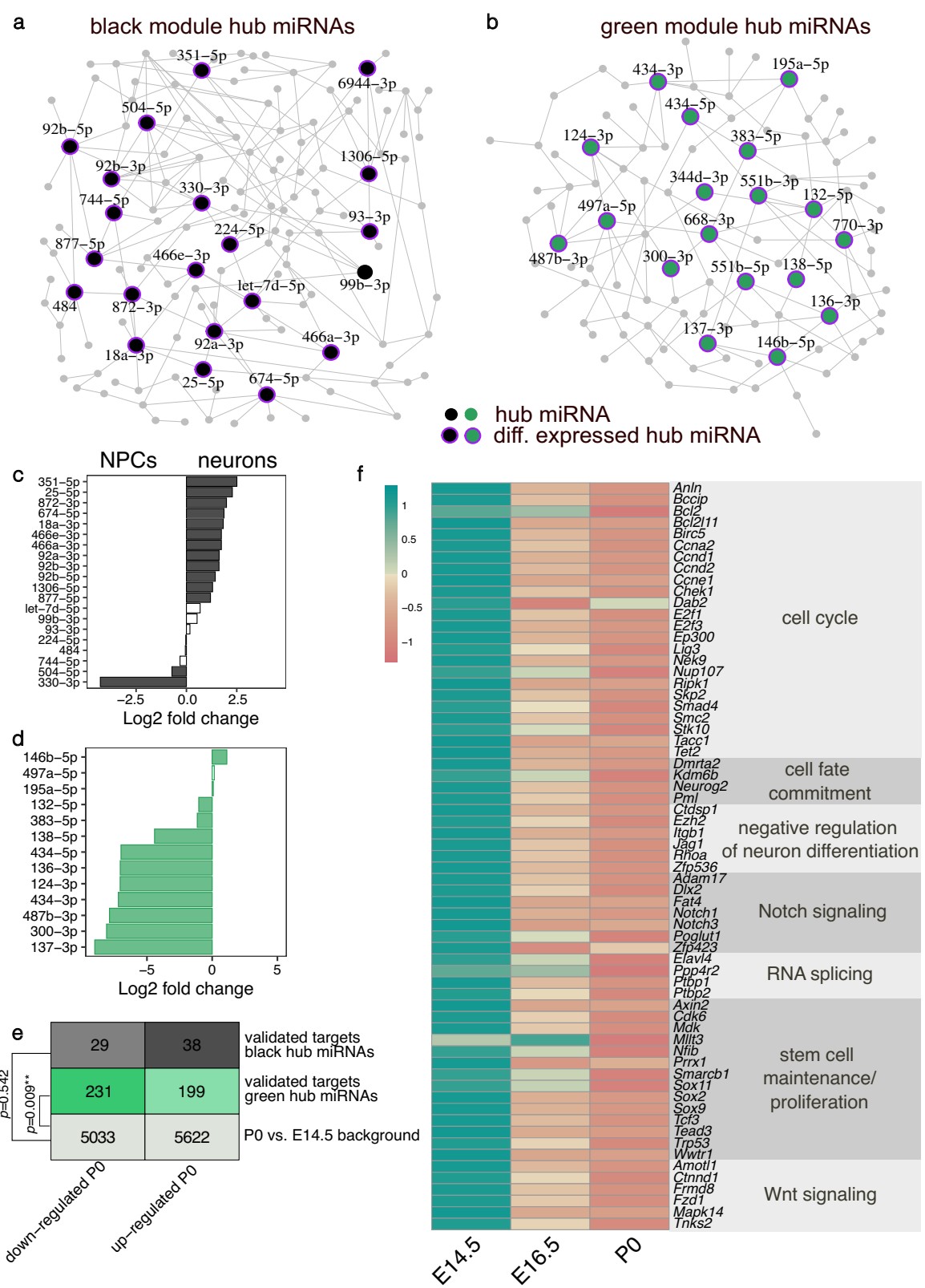

the black and green modules and identified key driver (hub) miRNAs as described previously[34,35]. The black module contained 21 hub miRNAs. 12 miRNAs were also upregulated in NPCs compared to neurons including members of the miR-92 family - miR-92a, miR-92b (Fig. 4c). Moreover, we detected 18 key driver miRNAs in the green module. Ten of these were also significantly upregulated in neurons compared to NPCs, including the

neuron-specific miR-124 and miR-137 (Fig. 4d). This analysis confirmed the stage-specific regulation of a core set of miRNAs during embryonic cortical development.

Importantly, all hub miRNAs in the green module and all but one miRNA in the black module were significantly differentially regulated at E14 compared to the P0 developmental stage (Fig. 4a, b). Furthermore, hub

**Fig. 4 | Hub miRNAs in the black and green WGCNA modules.** Network plot showing the hub miRNAs in the black (**a**) and green (**b**) modules. Hub miRNAs are represented as bigger nodes. Purple halos around the nodes indicate miRNAs that are significantly differentially expressed between E14 and P0. Differential expression of hub miRNAs from the black (**c**) and green (**d**) module detected in the NPCs versus neurons analysis. Positive log2 fold changes indicate an upregulation in NPCs, negative values correspond to upregulation in neurons. White bars indicate non-significant fold changes. **e** Distribution of differentially expressed genes in the mouse cerebral cortex between E14.5 and P0. The overall number of genes as well as the number of these genes that are validated targets from miRNAs in the black and green module as obtained from miRTarBase are given. The comparison of the distribution of differentially expressed targets of the black or green module from the overall distribution of differentially expressed genes was performed with a Fisher's exact test. **f** Z-score transformed average expression of selected validated targets of miRNAs from the green module that are significantly upregulated at E14.5 compared to P0. The expression values in (**e**) and (**f**) were obtained from the bulk RNA-seq of the mouse cerebral cortex across development study by Weyn-Vanhentenryck et al.[37].

miRNAs in both the black and the green modules showed significantly stronger module membership and gene trait significance estimates compared to the remaining miRNAs in the respective module (Supplementary Fig. 3e–h), thereby confirming their importance. Here, module membership corresponds to the correlation between the miRNA's expression and the module eigengene. Furthermore, the gene trait significance represents the correlation between miRNA expression and the trait of interest, in our study —the developmental stage[36].

To gain an insight into the relevance of hub miRNAs in regulating target gene expression across development, we re-analyzed bulk RNA-seq data from the mouse cortex at E14.5, E16.5, and P0 from the study by Weyn-Vanhentenryck and colleagues[37]. We limited this analysis to the experimentally validated targets of the conserved hub miRNAs that we obtained from miRTarBase[38]. We did not observe a significant deviation in the differential expression of target genes of the black module when comparing P0 and E14.5 (Fig. 4e). This could likely be explained by the overall lower number of validated targets for conserved key driver miRNAs from the black module (Supplementary Fig. 3d). However, the number of downregulated targets of the green module's hub miRNAs was significantly higher compared to the overall background distribution of differentially expressed genes (Fig. 4e). The increased proportion of genes with a reduced expression at the time point when the green module's hub miRNAs that target them have their expression peak strongly suggests that these miRNAs are involved in repressing these targets' expression in vivo.

To gain insight into the biological processes that the downregulated targets of green module hub miRNAs are involved in, we performed a GO annotation analysis (Fig. 4f). Notably, many of these genes are associated with the cell cycle, stem cell maintenance and proliferation (e.g., the transcription factors *Sox2, Sox9* and *Sox11*), cell fate commitment (e.g., *Neurog2*), and negative regulation of neuronal differentiation. Furthermore, several targets are involved in activating the Notch signaling pathway (e.g., *Notch1, Notch3*) and negatively regulating the Wnt pathway. Important mediators of RNA splicing (e.g., the RNA binding proteins *Ptbp1* and *Ptbp2)* were also among the downregulated targets. Notably, the green module hub miR-124-3p is known to repress *Ptbp1* expression and thereby induce neuron-specific alternative splicing programs required for neuronal differentiation[39]. Therefore, these results indicate that miRNAs might be particularly important for the switch from undifferentiated NPCs to differentiated neurons by silencing the expression of key factors that promote stem cell maintenance, proliferation and non-neuronal splicing patterns.

## Prediction of miRNA co-targeting networks in the developing cerebral cortex

WGCNA revealed that clusters containing multiple miRNAs are co-expressed during cortical development. Furthermore, when looking at the predicted targets of the black and green modules, we observed that only ~30% were potentially targeted merely by a single miRNA. On average, each gene was associated with more than 3 miRNAs (Supplementary Fig. 3a, b). This apparent redundancy of simultaneously expressed miRNAs that putatively bind to overlapping sets of genes supports the previously proposed hypothesis that miRNAs might act together to co-operatively exert stronger gene silencing (Fig. 1). To construct a comprehensive co-targeting network of miRNAs in the developing cerebral cortex, we designed a statistical framework that allowed us to detect which miRNAs share significantly more targets than expected by chance. In this analysis, we included all miRNAs whose module eigengene was correlated significantly with developmental time in the WGCNA (Fig. 3b). Furthermore, we only considered conserved miRNAs with at least 300 conserved targets and distinct seed sequences[20]. miRNAs belonging to the same broad conserved family and thereby having identical seed sequences, were grouped together for the statistical testing. After applying these filtering criteria, we performed the co-targeting prediction with a set of 77 miRNA families corresponding to 106 individual miRNAs (Fig. 5a). We detected 1216 significant co-targeting pairs with an adjusted *p*-value < 0.05 (Fig. 6, Supplementary data 4) and each miRNA had on average 41 co-targeting relationships (Fig. 5b). Interestingly, the number of significant co-targeting relationships was positively correlated with the miRNA's average expression level (Fig. 5c). This finding implies that important miRNAs might need to be produced at higher amounts to facilitate their involvement in multiple cooperative gene silencing interactions in the co-targeting network. Importantly, hub miRNAs such as miR-92a/b from the black module as well as miR-124 and miR-137 from the green module were among the miRNAs with the highest expression level and highest number of co-targeting relationships (Fig. 5c) indicating that these miRNAs might be master regulators in both co-expression and multi-targeting networks. As expected, the length of the 3' UTR was positively correlated with the number of significant miRNA pairs targeting a gene (Fig. 5d). Interestingly, *Nova1*, was among the top 10 genes involved in the highest number of co-targeting associations. NOVA1 is a neuron-specific splice factor that is crucial for neuronal viability[40] and regulates the alternative splicing of genes involved in synapse formation[41]. In line with this, we detected a significant *Nova1* expression increase in cortical samples at P0 compared to E14.5 (log2 fold change = 0.84, adj. *p*-value < 0.0001) when re-analyzing the RNA-seq data from Weyn-Vanhentenryck et al.[37]. Thus, our findings further highlight the potential importance of miRNAs in regulating neurodevelopment by influencing the expression of crucial effectors of alternative splicing processes.

Remarkably, we detected the strongest co-targeting relationship between miR-137 and the miR-25/miR-363/miR-92 family, sharing 353 common gene targets (Fig. 5f). While miR-137 was upregulated at later developmental stages, all members of the miR-25/miR-363/miR-92 cluster were regulated in the opposite direction. This prominent association between hub miRNAs, that were not co-expressed, prompted us to investigate the distribution of intra-modular (between miRNAs from the same module) and inter-modular (miRNAs from different modules) co-targeting relationships for the black and green WGCNA modules. While we expected that relationships between co-expressed miRNAs might be enriched, we indeed observed a comparable distribution of intra- and inter-modular co-targeting pairs (Fig. 5e). These findings point to two distinct potential regulatory mechanisms justifying the need for miRNAs to share significantly more targets than expected by chance. On the one hand, miRNAs with common targets regulated in opposite directions during cortical development might arise from the need to facilitate target gene regulation at different time points or in distinct cell types. On the other hand, co-expressed miRNAs that regulate the same genes could cooperatively bind to their targets to exert a stronger repressive effect compared to single miRNAs.

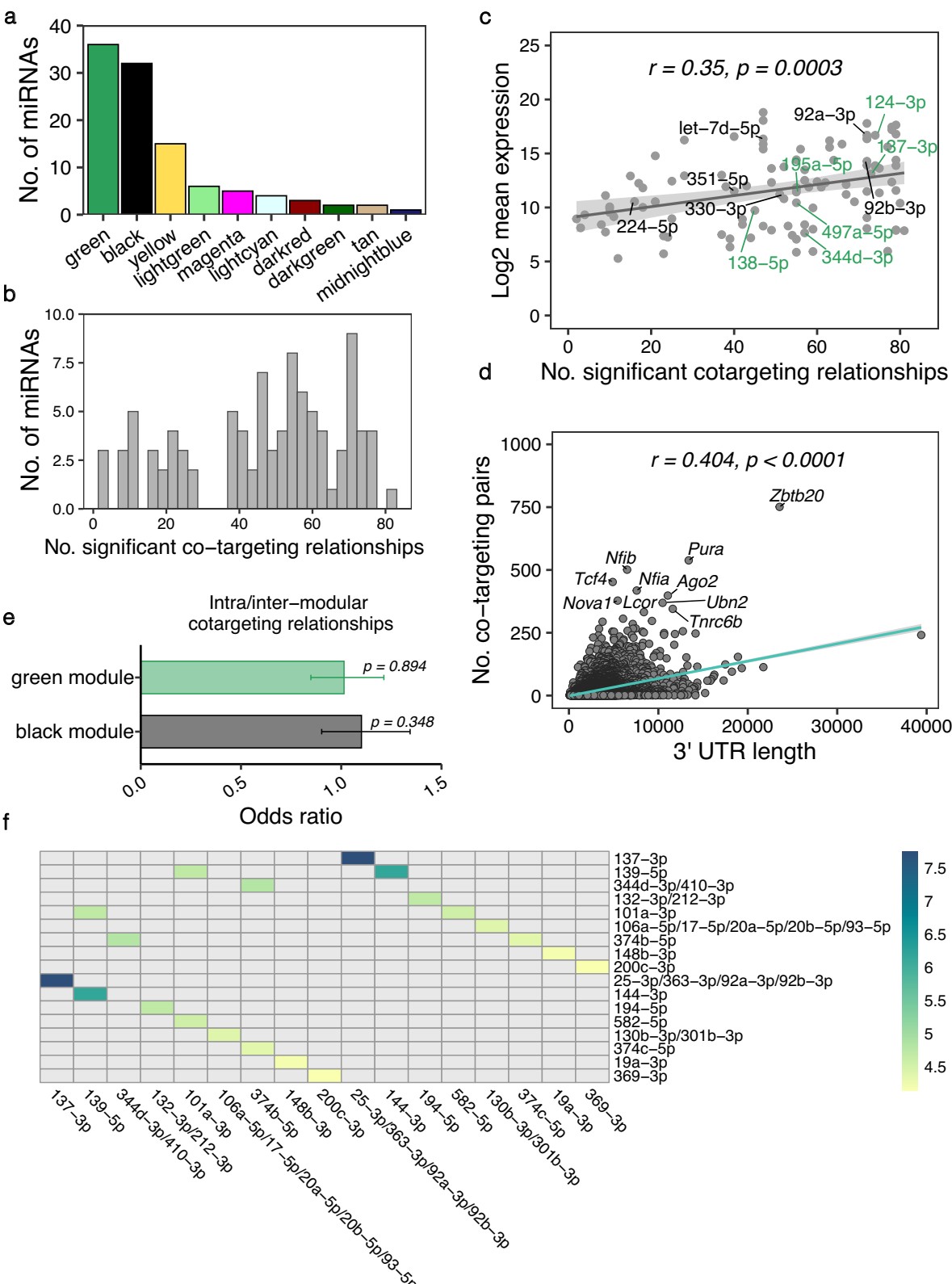

### Enhanced gene silencing effect by cooperative binding of co-expressed miRNAs

To validate the hypothesis that co-expressed miRNAs bind co-operatively to their common targets to enhance their repressive effect, we employed luciferase reporter assays focusing on miRNAs that were upregulated during embryonic brain development. We performed a gene ontology term look-up of all targets and filtered potential candidate genes for terms including brain/forebrain development and (central) nervous system development. Then we looked for genes with two or more conserved binding sites in their 3' UTR for miRNAs upregulated during cortical development. Using this strategy, we compiled a list of 7 candidate target genes (*Neurod1, Apc, Dcx, Ndst1,*

**Fig. 5 | Features of the miRNA co-targeting network of the developing cerebral cortex.** The co-targeting network was constructed using miRNAs expressed in the mouse cerebral cortex at E14, E17 and P0. **a** Bar plot showing the number of miRNAs from each WGCNA module that were included in the co-targeting analysis after applying the filtering criteria. **b** Histogram of the number of miRNAs having a specific number of significant co-targeting relationships. **c** Scatter plot indicating the correlation between the number of co-targeting relationships with the log2-transformed miRNA's average expression level; *r* corresponds to Pearson's correlation coefficient. Labels indicate hub miRNAs from the WGCNA analysis of the black and green modules **d** Correlation between the 3' UTR length of target genes and the number of significant co-targeting miRNA pairs the genes are targeted by; *r* corresponds to Pearson's correlation coefficient. **e** Bar plot with the association of the number of intra-modular (between miRNAs of the same module) and inter-modular (involving miRNAs from different modules) co-targeting relationships. Odds ratios were obtained with a Fisher's exact test. Error bars indicate the 95% confidence interval. **f** Heatmap with the top 10 strongest co-targeting relationships between pairs of miRNAs/conserved miRNA families. Values indicate odds ratios as measures of association between the co-targeting pairs.

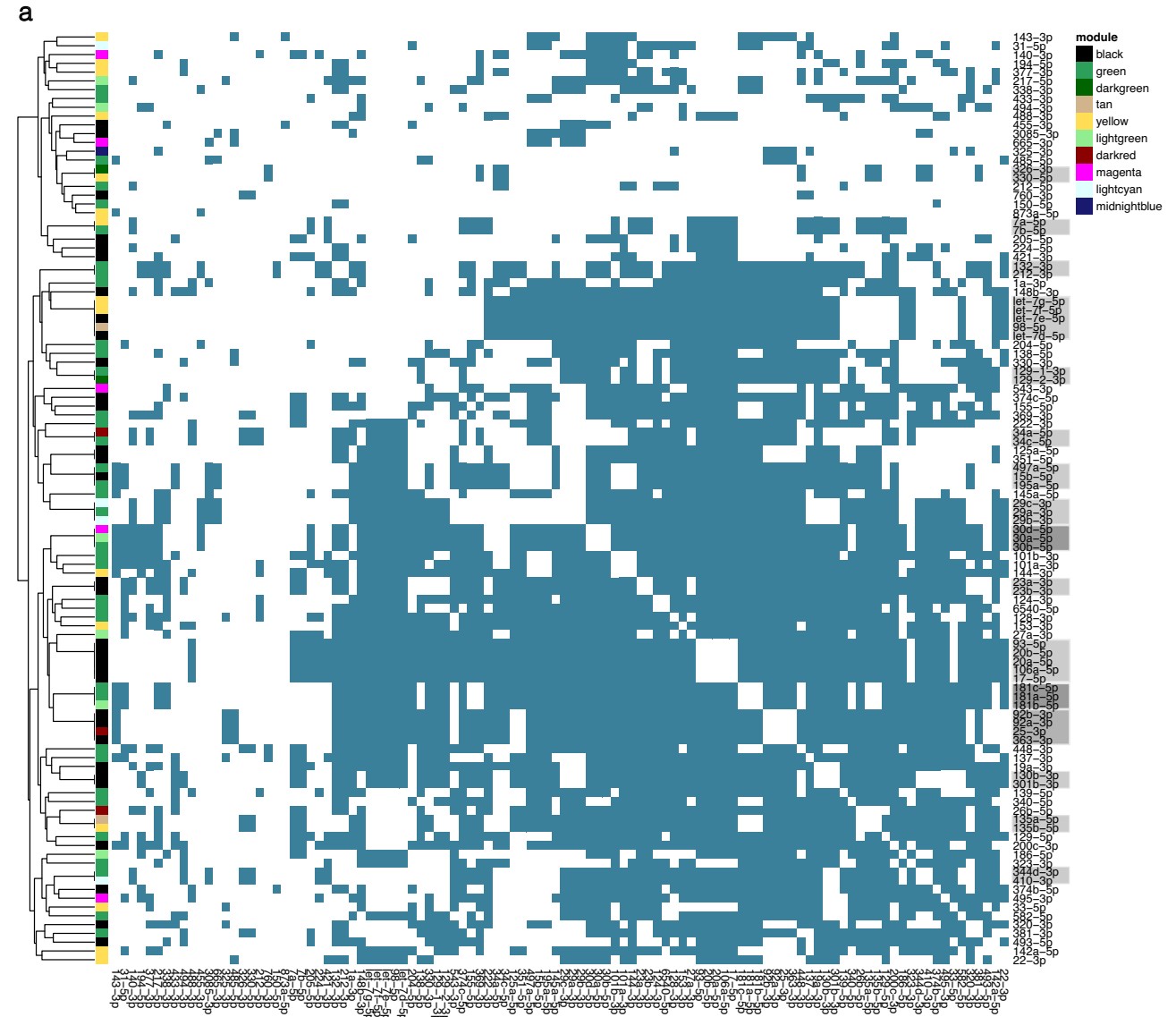

**Fig. 6 | Significant co-targeting miRNA pairs in the mouse cerebral cortex.** Colored squares on the heatmap indicate significant co-targeting relationships. miRNAs that belong to the same conserved family and therefore have identical co-targeting relationships to each other, are indicated with gray rectangles on the vertical axis of the heatmap.

*Zeb2, Src* and *Cxcl12*) and 5 miRNAs (miR-128-3p, miR-129-5p, miR-135b-5p, miR-137-3p and miR-153-3p).

First, we performed RT-qPCR analyses to validate the expression patterns that we observed in the bulk sequencing of the miRNAs selected for luciferase assays. All 5 miRNAs showed significantly increased expression levels in neurons compared to NPCs with mean relative quantification (RQ) values ranging from 2.9 to 3987 in the case of miR-135b-5p (Supplementary Fig. 4a–e). Furthermore, all miRNAs were associated with increased expression at later developmental stages in cortical tissue samples. The

relative expression of miR-128-3p and miR-135b-5p significantly increased over all time points analyzed (Supplementary Fig. 4f, h). miR-137-3p and miR-153-3p were associated with a significantly higher expression levels at E17 and P0 compared to E14 (Supplementary Fig. 4 i, j). miR-129-5p was the only miRNA that showed a non-significant trend of higher expression at later developmental stages (Supplementary Fig. 4g).

Subsequently, for each luciferase reporter assay, we cloned the target gene's 3'UTR fragment containing the binding sites of the selected miRNAs, downstream of the synthetic *Renilla* luciferase gene into the psiCHECK™-2

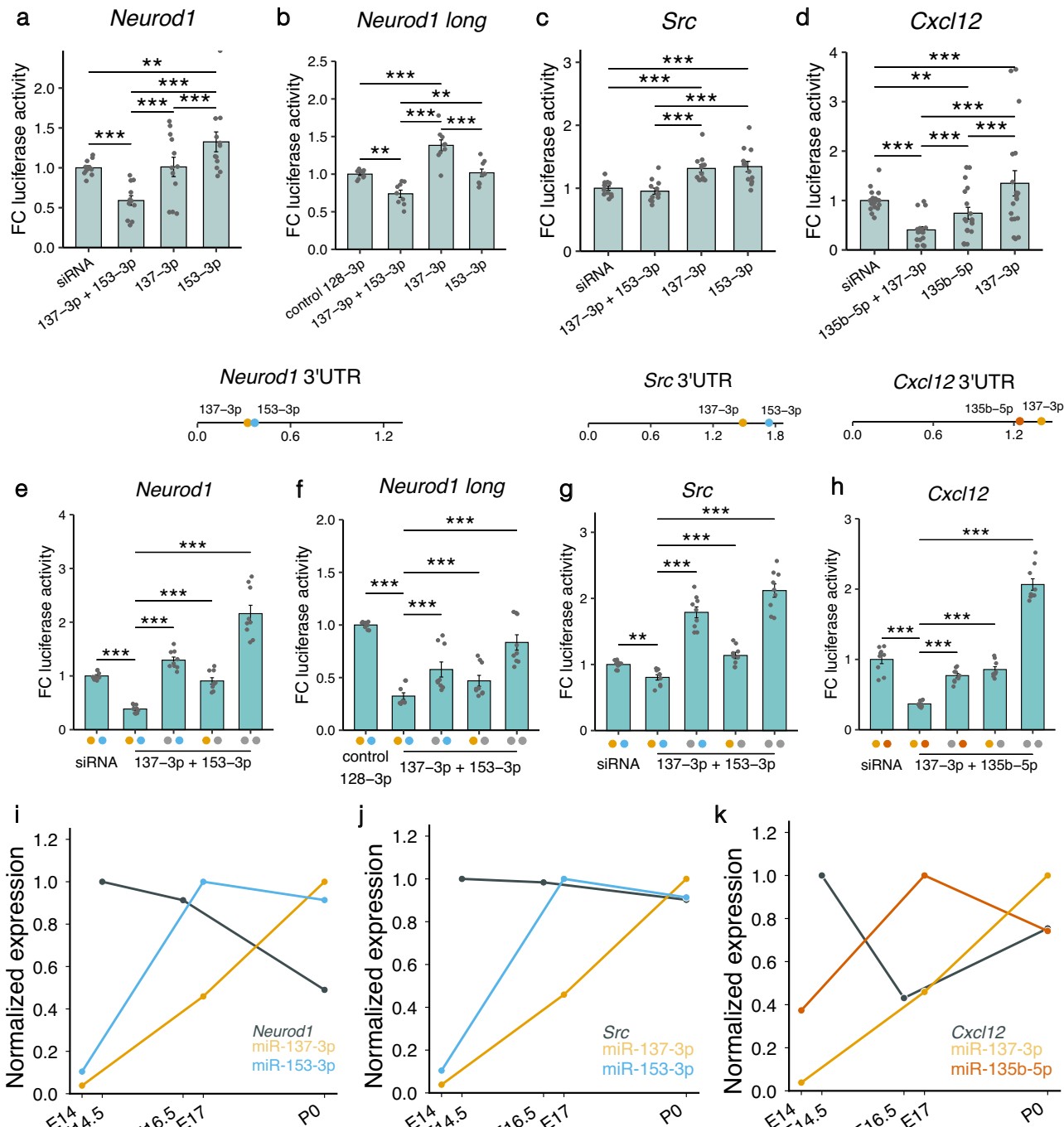

**Fig. 7 | Luciferase activity in lysates of HEK293 cells transfected with plasmids containing 3' UTR fragments of target genes. a–d** Lysates were co-transfected with different combinations of miRNA mimics. Fold change (FC) of luciferase activity was obtained by calculating the ratio of *Renilla* luciferase and firefly luciferase activity and then normalizing to the mean of the control siRNA or miRNA group. Locations of the binding sites in the 3' UTR of the respective gene are represented by colored dots. The length of the 3' UTRs is indicated in kbp. **e–h** Fold change of luciferase activity was measured after introducing point mutations in the binding sites of the miRNAs in the 3' UTR regions. Colored dots on the x-axis indicate that the binding site of the respective miRNA was intact, gray dots indicate a mutated binding site. **i–k** Expression of miRNAs and target genes during embryonic brain development. Expression values were normalized to a 0–1 range. miRNA expression was quantified by bulk RNA-seq of the mouse cerebral cortex at E14, E17 P0. Target gene expression values were measured at E14.5, E16.5 and P0 and were obtained from the bulk RNA-seq study of the mouse cerebral cortex at different developmental stages by Weyn-Vanhentenryck et al.[37]. Data are shown as mean ± standard error of the mean and individual values. *$p < 0.05$, **$p < 0.01$, ***$p < 0.001$, two-way analysis of variance followed by Tukey's post-hoc test. $N = 12$ replicates in (**a**) and (**c**), $n = 18$ in d, $n = 9$ in (**b**), (**c**) and (**e–h**), $n = 2$ in (**i–k**).

vector, which also expresses firefly luciferase as internal control. Plasmids were then co-transfected into HEK293 cells with different combinations of miRNA mimics and luciferase activity was evaluated in cell lysates 48 h later. The fold change of *Renilla*/firefly luciferase signal was normalized to a control group containing either a non-targeting siRNA or a miRNA

mimic without a predicted binding site in the 3'UTR fragment of the target gene.

The first gene we investigated, *Neurod1*, is an important transcription factor regulating neuronal differentiation and migration in the developing cerebral cortex[42]. *Neurod1* contains binding sites for miR-137-3p and miR-

153-3p that were previously demonstrated to co-operatively repress the gene's expression[20]. Therefore, we used this experiment as a positive control. We independently cloned two *Neurod1* 3' UTR fragments with different lengths (274 bp and 1188 bp, respectively). The luciferase activity was significantly reduced in cells transfected with the combination of miR-137-3p and miR-153-3p compared to individual miRNAs as well as two different negative controls (siRNA and miR-128-3p), thereby confirming the co-targeting effect previously observed (Fig. 7a, b).

*Src*, a proto-oncogene that codes for a non-receptor protein tyrosine kinase is another important neurodevelopmental regulator that contains binding sites for both miR-137-3p and miR-153-3p in its 3'UTR. *Src* is expressed at steadily high levels in the mouse neocortex from E12.5 to P1 and overexpression of this gene leads to impaired neuronal migration due to altered adhesion properties and cytoskeletal dynamics[43]. We observed that the combination of both miRNAs exerted a significantly stronger repressive effect on *Src* compared to each individual miRNA (Fig. 7c).

As an additional target gene for the reporter assays, we selected *Cxcl12* which is also an important regulator of early brain development. *Cxcl12* is involved in migration of NPCs, early localization of Cajal-Retzius cells in the developing cortex, and in axon guidance and pathfinding[44,45]. *Cxcl12* is a predicted target of miR-137-3p and miR-135b-5p. Luciferase reporter assays showed that co-transfection with both miRNAs significantly increased the repressive effect compared to the negative siRNA control and treatment with each miRNA separately (Fig. 7d).

In contrast, we did not observe a significant co-targeting effect for miR-153-3p with miR-129-5p and miR-137-3p with miR-128-3p in reporter assays with *Apc* and *Ndst1*, respectively (Supplementary Fig. 5a, c). Furthermore, we aimed to investigate if higher-order co-targeting interactions, including three miRNAs would exert stronger silencing effects on gene expression. To this end, we co-transfected cells with plasmids containing the 3' UTR fragment of the *Dcx* gene and mimics of miR-128-3p, miR-129-5p, and miR-135b-5p (Supplementary Fig. 5b). In an additional assay, we cloned the 3' UTR of *Zeb2* and co-transfected cells with its targeting miRNAs – miR-129-5p, miR-137-3p and miR-153-3p (Supplementary Fig. 5d). Treatment with different miRNA mimics significantly reduced luciferase activity compared to the siRNA control in the *Zeb2* assays. However, we did not observe a significant cooperative repressive effect for either *Dcx* or *Zeb2*.

To confirm that the cooperative silencing effect for *Neurod1, Src,* and *Cxcl12* was mediated via direct binding of the miRNAs, we employed in vitro site-directed mutagenesis to introduce mutations in the miRNA binding sites of the 3' UTRs. We generated plasmids carrying a mutation in one of the binding sites or in both binding sites of the respective co-targeting pair. For all three genes, we observed the highest luciferase activity in the cell lysates where both miRNA binding sites were mutated. Conversely, the strongest repressive effect was detected when both binding sites were intact, thereby confirming that miRNAs acted co-operatively to exert stronger gene silencing (Fig. 7e–h). Furthermore, we investigated the expression patterns of these three genes by employing the data from Weyn-Vanhentenryck et al.[37]. *Neurod1* was significantly downregulated at P0 compared to E14.5 and E16.5, correlating negatively with the expression pattern of the co-targeting pair miR-137-3p and miR-153-3p (Fig. 7i, Supplementary Fig. 6a). While *Src* was not differentially regulated, it still had its expression minimum at P0 (Fig. 7j, Supplementary Fig. 6b). Interestingly, *Cxcl12* had a significantly reduced expression only at E16.5 compared to E14.5, but this pattern negatively correlated with the expression peak of miR-135b-5p at E17 (Fig. 7k, Supplementary Fig. 6c). These matching in vivo expression patterns of the miRNAs and the genes they co-target lend further support to the biological relevance of the co-operative miRNA gene silencing we observed in the luciferase assays.

## Discussion

Our study provides a detailed map of the longitudinal changes in miRNA expression patterns that occur in the transition between NPCs and neurons as well as in vivo at key stages of embryonic cortical development in the

mouse. While previous studies have already employed high-throughput sequencing to examine miRNA composition in the embryonic cortex, they have focused on a single time point only[29,46]. In contrast, in situ hybridization-based technologies have successfully profiled miRNA expression in different embryonic brain structures and at multiple time points[47], however, this analysis has been limited to a pre-selected set of miRNAs. To our knowledge, this is the first high-throughput longitudinal investigation, therefore providing a valuable resource for elucidating the regulatory role miRNAs play during murine embryonic corticogenesis.

Remarkably, the majority of the miRNAs significantly altered their expression level already at the transition from E14 to E17. This predominantly neurogenic phase is characterized by a complex interplay of multiple processes that shape late-stage embryonic corticogenesis[48]. At E14, radial glia cells in the ventricular zone have switched from symmetrical to asymmetrical divisions to give rise to the basal progenitors that later generate neurons. Subsequently, newly born neurons migrate out of the ventricular zone to the upper layers of the neocortex where they differentiate into mature neurons. Upon neurogenesis completion at around E17, neural stem cells give rise to the other brain cell types including astrocytes and oligodendrocytes[28,30]. Cell fate commitment and neuronal identity are determined by specialized transcriptional programs that occur in sequential waves and eventually result in the immense cell type diversity observed in the brain[49,50]. The dynamic changes of miRNA expression we detected indicate that miRNA networks can act as quick regulators of these transcriptional programs by controlling the expression of cell-type specific genes[51]. miRNA-mediated transcriptional inhibition could also contribute to reducing excess levels of target mRNA arising as noise from transcriptional bursts and thereby ensuring appropriate cell-type or developmental-stage specific expression.

By employing a co-expression network analysis, we detected sets of co-regulated miRNAs with individual miRNAs acting as hubs at early (e.g., miR-92a/b) and late (e.g., miR-124 and miR-137) stages of neurogenesis. miR-92 is a part of the miR-17-92 cluster that is involved in the maintenance of neural stem cells, and its genetic ablation results in a reduced pool of neural stem and radial glia cells and a premature transition to intermediate progenitors[52]. Furthermore, miR-137 has been reported previously to influence neuronal differentiation via a regulatory loop between TLX and miR-137' downstream target LSD1[53].

Interestingly, we observed an enrichment of downregulated genes at P0 compared to E14, which were validated targets of hub miRNAs in the green module (Fig. 4e). This module consisted of miRNAs that were upregulated at P0. A GO annotation analysis revealed that many of these genes are regulators of key processes that are responsible for keeping neural stem cells in a proliferative state and blocking neuronal differentiation. For instance, the canonical NPC marker *Sox2* is crucial for the maintenance and self-renewal of progenitors[54,55], and silencing its expression promotes neuronal differentiation[54]. Another member of the SOX transcription factor family— *Sox9*, also plays an important role in maintaining neural stem cells in the early embryonic neocortex and its expression levels determine self-renewal and neurogenic division behavior of radial glial cells[56]. In fact, a recent review highlighted a complex regulatory interplay between SOX transcription factors and miRNAs that guide distinct cellular activities in the developing and adult brain under physiological and pathological conditions[57].

Green module hub miRNA targets that were downregulated at P0 also included positive regulators of Notch signaling and negative regulators of the Wnt pathway. Activation of the Notch pathway indeed leads to maintaining the neural progenitor pool[58]. Furthermore, inhibiting Wnt signaling during mid and late stages of neurogenesis in the neocortex was previously reported to result in reduced neuronal production[59].

Another crucial mechanism through which miRNAs can influence the development of the cerebral cortex is via the regulation of RNA-binding proteins. In line with this, we observed that several effectors of alternative splicing whose expression significantly changed during embryonic corticogenesis were validated targets of hub miRNAs (Fig. 4f). Alternative splicing plays a pivotal role in neuronal differentiation[60], axonogenesis[61] and

synapse formation[62]. In this context, miR-124 represents one of the most well-studied examples in neuronal differentiation. miR-124 targets PTBP1, a protein that represses neuronal splice patterns. The upregulation of miR-124 in neurons is sufficient to suppress the expression of *Ptbp1* and thereby induce the necessary splicing changes for the transition from NPCs to neurons[39]. Thus, miRNAs could act as a post-transcriptional mechanism ensuring the precise spatio-temporal expression of RNA-binding proteins which is critical for proper development of the central nervous system[63].

Surprisingly, the gene silencing effect of most miRNAs is modest despite their crucial regulatory potential[64], indicating that—with the exception of hub miRNAs—they might be fine tuners rather than master regulators of embryonic brain development. However, the redundancy of multiple co-expressed miRNAs sharing the same target genes supports the mechanism of co-operative binding to enhance the gene-silencing effect of individual miRNAs[20]. Accordingly, we confirmed the co-targeting for several neurodevelopmentally relevant genes in luciferase assays (Fig. 7). Furthermore, the negative correlation between the in vivo miRNA and target expression peaks strongly suggests that co-targeting is a biological phenomenon that can occur during embryonic cortical development. As 3' UTRs of mRNAs expressed in the brain are longer compared to other tissues[65] and this length even increases in neuronal transcripts[66], the brain indeed offers extremely favorable conditions for the emergence of miRNA multi-targeting networks[20].

Notably, significant co-targeting relationships between co-expressed miRNAs were as likely as co-targeting associations between miRNAs regulated in opposite directions. Apart from simultaneous cooperative biding to enhance gene silencing, this observation points to an alternative evolutionary need for the presence of multiple binding sites for distinct miRNAs in the same gene, namely regulating the expression at different time points or in diverse cell types. Accordingly, Nowakowski and colleagues recently reported cell-type specific miRNA-mRNA interactions in the developing human cortex using a single-cell qPCR profiling strategy[51]. Thus, distinct miRNAs might be responsible for silencing temporally or spatially abnormal expression of an overlapping set of genes. This explains our observation of miRNAs with opposite expression patterns that share more targets than expected by chance.

In summary, we detected dynamic changes in miRNA expression during embryonic brain development with distinct miRNAs acting as hubs in co-expression and co-targeting networks. Furthermore, we showed that miRNAs might be particularly important for controlling cell fate commitment and neuronal differentiation by silencing the expression of genes that promote neural stem cell proliferation and maintenance as well as NPC-like splicing patterns. Our study also provides additional evidence that simultaneous binding to common targets increases the transcriptional repression effect of miRNAs. To further resolve the complexity of mRNA-miRNA as well as miRNA co-targeting networks in distinct cell types in vivo, future studies should focus on single-cell high-throughput profiling techniques.

## Methods
### miRNA sequencing of the developing mouse cerebral cortex and in NPCs/neurons
For bulk RNA sequencing of cerebral cortex samples as E14, E17 and P0, the two cortical hemispheres of each embryo were dissected from the brains, the meninges were removed, and the cortices were stored in RNAlater solution (Sigma). To isolate total RNA (including miRNAs) from the embryonic cortices, the Trizol/Chloroform method was used. For NPC/neuron culture, the cortices of E14.5 NMRI embryos (Janvier Labs; Le Genest-Saint-Isle, France) were dissected and collected in cooled DMEM high glucose media (Gibco Life Technologies) and processed into single cells by trypsin digestion. For NPC and neuron culture, $0.2 \times 10^6$ cells per well and $1.4 \times 10^6$ cells per well, respectively, were seeded on Poly-L-ornithin and Laminin coated 6 well plates. The NPCs were cultured in neurobasal medium containing 2% B27 -VitA supplement (Gibco Life Technologies), 500 μM Glutamax (Gibco Life Technologies) and EGF (10 ng/ml) and FGF (10 ng/ml). For neural

differentiation, the cells were cultured in neurobasal medium containing 2% B27 supplement (Gibco Life Technologies) and 500 μM Glutamax (Gibco Life Technologies). Total RNA, including miRNA, was isolated from the NPCs and neurons as well as from cortices of female and male NMRI embryos (Janvier Labs; Le Genest-Saint-Isle, France) at the time points E14, E17 and P0 ($n = 6$ embryos per sex and time point) using the Qiagen miRNeasy mini kit (CatNo. 217004; Qiagen; Hilden, Germany). To determine the concentration of the isolated RNA a NanodropOne Spectrometer was used (ThermoScientific). 500 ng of total RNA, including miRNA, were used as an input for library preparation with the Bioo Scientific NextFlex small RNA v3 Seq Kit (CatNo. NOVA-5132-05; Bioo Scientific; Austin, USA). Library preparation was conducted according to the kit manual. Size distribution and concentrations of the prepared libraries were checked by Qubit dsDNA High Sensitivity Assay (CatNo. Q32851; Thermo Scientific; Waltham, USA). 0.5 nanomoles (NPCs/neurons) or 4 nanomoles (cerebral cortex samples) of the prepared miRNA libraries were loaded on a High Output v2 kit (75 cycles) Illumina cartridge which was run on a NextSeq 500 device.

### miRNA sequencing data pre-processing
After the sequencing, bcl2fastq v2.17.1.14 conversion software (Illumina, Inc.) was used to demultiplex sequencing data and convert base call (BCL) files into fastq files. The trimming of the fastq files was conducted in two steps as suggested by the NEXTflex™ Small RNA Trimming Instructions. Briefly, sequencing adapters (TGGAATTCTCGGGTGCCAAGG) were trimmed and reads shorter than 15 nucleotides were removed from further analysis. Afterwards, 4 bases from either side of each read were trimmed using Cutadapt v1.18. Quality control checks were performed on the trimmed data with FastQC v0.11.7.

### miNRA differential expression analysis
miRNA samples with at least 20 million reads were further analyzed, thus 2 female samples at E14 not meeting this criteria were excluded. Using miRDeep's v2.0.1.2 *mapper.pl* script, miRNA reads were mapped to the *Mus musculus* GRCm38 genome. Afterwards, known and novel miRNA were identified using the *miRDeep2.pl* script. Prior to differential expression analysis, miRNAs with CPM expression values less than 10 were filtered out using edgeR v3.30.3. Differential expression analysis was then performed with DESeq2 v1.28.1. miRNAs with an adjusted $p$-value < 0.05 and an absolute log fold change exceeding 0.5 were considered differentially expressed. $P$-values were corrected for multiple comparisons with the Benjamini–Hochberg method.

### Weighted gene co-expression network analysis
We performed a Weighted Correlation Network Analysis (WGCNA)[36] to find clusters or "modules" of co-expressed miRNAs. For the analysis, a *bicor* correlation type and a *signed* network type with soft-thresholding power of 24, minimal module size of 5 and dynamic tree cut height of 0.2 were applied. Target prediction of the miRNAs in each module was performed using the TargetScanMouse database v7.2[67]. Functional enrichment analysis of the different modules in the co-expression network was conducted using the clusterProfiler v4.6.2 R package and the target genes of the miRNAs in the respective module as input.

### Key driver analysis
To identify hub (or key driver) miRNAs in the modules of the co-expression network, we performed a key driver analysis as previously described[34,35]. First, we used the algorithm for the reconstruction of cellular networks (ARACNE)[68] as implemented in the bnlearn R package v4.8.3 to obtain an undirected gene regulatory network from the expression levels of the miRNAs in the respective WGCNA module. Next, we performed key driver analysis on the ARACNE reconstructed network using the KDA R package v0.2.2 (https://github.com/mw201608/mnml-public/tree/master/pkgs) to identify important regulatory miRNAs.

## Analysis of publicly available bulk RNA-seq data of the developing mouse cortex

To explore the expression of target genes during cortical development, we re-analyzed publicly available data from the study by Weyn-Vanhentenryck et al.[37] containing RNA-seq of the mouse cortex. We focused on E14.5, E16.5, P0 as these were comparable with the time points in our study. Fastq files were downloaded from the NCBI Short Read Archive, accession number SRP055008. After trimming with BBDuk v39.01, reads were mapped to the Gencode mm39 reference genome (released 19.10.2022) using STAR v2.7.10b. A count matrix was obtained using FeatureCounts provided by SubRead v2.0.6. Differential expression analysis was performed with DESeq2 v1.40.1 with default settings. Genes with an adjusted p-value (Benjamini–Hochberg method) <0.05 were considered to be differentially expressed.

## miRNA expression profiling using RT-qPCR

miRNA expression patterns identified by small RNA sequencing were validated with the TaqMan MicroRNA Assay (Applied Biosystems). Reverse transcription was performed using the TaqMan MicroRNA Reverse Transcription Kit (Applied Biosystems; CatNo. 4366596). RT-qPCRs were conducted with the TaqMan Universal Master Mix II, no UNG (Applied Biosystems; CatNo. 4440040) on a StepOnePlus Real-Time PCR System (Applied Biosystems). The U6 snRNA was used as a control for normalization of experimental samples. Relative quantification (RQ) values of miRNA expression were calculated with the ΔΔCT method. Expression levels in NPCs and E14 samples, respectively, were used as reference groups. Primers used for the RT-qPCR analysis are shown in Supplementary Table 1.

## miRNA co-targeting prediction

miRNA co-targeting analysis was performed using a custom R script. We employed the TargetScanMouse database v7.2 to identify the broadly conserved mRNA targets for each miRNA. miRNAs which belong to the same broadly conserved family and therefore have identical gene target sets were grouped together for further analysis. Additionally, we only kept miRNAs with at least 300 targets[20].

Subsequently, we created a custom control set for each miRNA, containing genes with similar 3′ UTR length, GC content, and sequence conservation as the actual genes included in the respective target set based on a case-control strategy. To select the best matching control for a gene X from the respective target set (case), first we filtered the potential pool of candidates for genes with a 3′ UTR length within the range of 0.85–1.15 of the 3′UTR length of target gene X. Then, we filtered out genes with a GC content outside the range of 0.95–1.05 of the GC content of gene X. Finally, we eliminated candidates with a phyloP score outside the range of 0.8–1.2 for gene X. If more than one candidate remained after the filtering steps, a control gene Y for the target gene X was picked randomly. If the filtering steps returned an empty set, no control gene was selected for the respective target. This procedure was repeated for all genes included in the target set for the miRNA. The filtering cut-off values were selected empirically to maximize the number of controls while simultaneously minimizing the difference in the 3′ UTR length, GC content, and phyloP score distributions of genes in the target and control sets. The similarity of the distributions of the three parameters between the control and target sets was ensured by non-significant pairwise Wilcoxon tests.

We considered miRNA A and miRNA B to be a co-targeting pair if the intersection of their target sets contained more genes than what would be expected by chance. To this end, we compared the observed number of common targets with the expected number obtained by intersecting the target set of miRNA A with the control set of miRNA B and vice versa. This yields two comparisons for each pair of miRNAs whose significance was evaluated with a Fisher's exact test. If both tests were statistically significant after a false discovery rate p-value adjustment (Benjamini–Hochberg method), then we considered miRNA A and miRNA B to be a bidirectional co-targeting pair. To quantify the magnitude of the co-targeting relationship, we calculated the odds ratios with values significantly higher than 1

## Table 1 | Contingency table used to calculate the association between intra- and inter-modular miRNA co-targeting relationships for the black and green WGCNA modules

|  | Significant co-targeting relationship | Non-significant co-targeting relationship |
|---|---|---|
| Intra-modular | m | M–m |
| Inter-modular | n | N–n |

m represents the number of significant intra-modular co-targeting relationships, M corresponds to the total number of possible intra-modular relationships, n corresponds to the number of significant inter-modular relationships and N refers to the total number of possible inter-modular relationships. We used the following formula to calculate M:

$$M = x(x - 1)/2$$

where x corresponds to the number of miRNAs in the respective module. The total number of possible inter-modular relationships was estimated with the formula:

$$N = xy$$

with x being the number of miRNAs in the module of interest and y corresponding to the number of miRNAs in the remaining modules.

corresponding to a pair of miRNAs sharing a significantly higher number of targets than what would be expected by chance. Higher odds ratios indicate a higher number of shared targets. To ensure that co-targeting pairs do not originate from identical seed sequences, miRNAs having identical seeds up to one mismatch were excluded a priori from the statistical analysis.

## Comparing the number of intra- and inter-modular co-targeting relationships

To investigate if the black or green WGCNA modules contain more intramodular (miRNAs of the same module) than inter-modular (miRNAs from different modules) co-targeting relationships, we performed a Fisher's exact test on the contingency table shown in Table 1.

## Luciferase reporter assays

3′ UTR fragments of miRNA target genes (for primer sequences see Supplementary Table 2) were cloned into the XhoI/NotI sites downstream of the synthetic Renilla luciferase gene of the psiCHECK-2 vector (Promega; CatNo. C8021). Mutations in the miRNA binding sites were introduced using the QuikChange II XL system (Agilent Technologies; CatNo. 200521-5). Primers used are shown in Supplementary Table 2. 5 × 10⁴ HEK293 cells were seeded into each well of a 12-well plate. 24 h later, 200 ng psiCHECK-2 reporter plasmid were co-transfected with 5 μl miScript miRNA mimics (Qiagen; CatNo. 219600) using 4 μl Lipofectamine 2000 Transfection Reagent (Thermo Fisher Scientific; CatNo. 11668019). Luciferase reporter assays were conducted 48 h later using the Dual-Luciferase Reporter Assay System (Promega; CatNo. E1980). Luciferase activity was measured on a CentroXS LB 960 (Berthold Technologies).

## Statistics and Reproducibility

In the analysis of qPCR and luciferase reporter assay data, two groups were compared statistically using an unpaired t-test or Welch's t-test in case of unequal variance between groups. For comparisons of three groups, a one- or two-way analysis of variance (ANOVA) followed by Tukey's post hoc test was performed. The assumptions of the linear model were evaluated by inspecting Q-Q plots and fitted values vs. residuals plots. Relative quantification values from the RT-qPCR assays were log_e-transformed prior to statistical testing. All p-values are two-tailed and a p-value ≤0.05 was considered statistically significant. Statistical analysis was performed using R v. 4.2.2.

## Reporting summary

Further information on research design is available in the Nature Portfolio Reporting Summary linked to this article.

## Data availability

The raw RNA-seq data were uploaded to the Sequence Read Archive (SRA) data base under the accession number PRJNA1018560. Individual values underlying figures are provided in Supplementary data 5.

## Code availability

All analysis scripts can be obtained from the corresponding authors upon reasonable request.

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

## Acknowledgements
H.T. and S.G. acknowledge funding by the Landesinitiative Rheinland-Pfalz and the Resilience, Adaptation, and Longevity (ReALity) initiative of the Johannes Gutenberg University of Mainz. S.W. was funded by the Emergent Algorithmic Intelligence initiative of the Johannes Gutenberg University Mainz supported by the Carl-Zeiss foundation. J.W. acknowledges funding from the Deutsche Forschungsgemeinschaft (WI-3837 / 8-1).

## Author contributions
H.T., D.H., and S.W. performed the bioinformatics analysis. L.S. and H.M. performed miRNA sequencing, RT-qPCR and luciferase experiments. H.T. and J.W. wrote the manuscript. S.G. and J.W. supervised the study. All authors read and approved the final version of the manuscript.

## Funding

## Competing interests
The authors declare no competing interests.

## Ethics
We have complied with all relevant ethical regulations for animal use. Ethical review and approval were not required for the animal study because the study did not include any animal experiments requiring approval. To carry out this study, mice were killed for organ removal. In Germany, this procedure is notifiable but does not require approval by an ethics committee.
