## [Transparent Peer Review file · Communications Biology]

Point-by-point responses to reviewers.

Reviewer #1 (Remarks to the Author):

I would like to comment as an experimental researcher. This is an interesting study that provides a comprehensive analysis of the dynamics of miRNAs and their target mRNA networks altered during the progression of neural development. By re-analyzing previous reports and data, this study not only provides logical evidence for the cooperative function of multiple miRNAs, which had been expected from the results of individual miRNA studies, but also provides new findings and is a valuable resource for further study. The manuscript is written in an easy-to-understand manner, even for experimental researchers, and is considered sufficient for publication. Therefore, the following are suggestions, not requirements.

Reply: We thank the reviewer for the positive feedback and recognizing the relevance and added value of our work. We are especially grateful to see that the computational aspects of our manuscript are also logical and understandable for researchers with a primarily experimental focus. We have implemented the suggestions outlined below.

Minor points:

1. It would be helpful to clarify the type of analysis and cell source for each experiment, such as single cell or bulk RNA-seq, cultured cells or tissue-derived non-cultured cells, etc.

Reply: We have updated the figure descriptions and manuscript text to include more details on type of analysis and source material for each experiment (Fig. 1; lines 114; 164; 199-200; 225; 264; 322; 402-403).

2. It might be worth to mention what miRNAs are classified in modules other than black and green, such as yellow, red, and etc.

Reply: We have provided detailed information on the classification of miRNA into different modules in the WGCNA analysis including miRNA family, genes targeted by the respective miRNA according to the TargetScan data base as well as family conservation information in Supplementary Table 2. We have updated the manuscript to include this information (lines 178-179).

3. In addition to NPCs vs. neurons, neurogenic NPCs vs. gliogenic NPCs are also known to be important microRNA-regulated transition in neural development. In the case of bulk RNA-seq analysis, the results would show cell heterogeneity-dependent changes in miRNA expression profiles. If you performed the analysis by scRNA-seq data, you could also add the results of changes in miRNA expression profiles associated with changes in neurogenic-to-gliogenic NPCs. In this neurogenic-to-gliogenic transition of NPCs, several reports have shown that the properties of NPCs alter between E14 and E15.

Reply: The reviewer raises a very interesting point as the same neural precursor cells that initially give rise to neuronal lineages in the developing brain later switch to glial cell differentiation. miRNAs could indeed play an important role in the switch from neurogenic-to-gliogenic differentiation competence. Since we performed bulk sequencing experiments and high-throughput profiling of miRNA expression on the single cell level is experimentally rather challenging, our analysis indeed likely reflects heterogenous miRNA expression profiles. It is however worthy to mention that several of the developmental-stage dependent expression changes we observed fit well with previously reported roles of miRNAs in neuro- and gliogenesis, respectively. For instance, miR-106a-5p and its paralog miR-17 were up-regulated at E14 in our analysis and expression levels were significantly reduced at E17/P0. Both miRNAs were previously reported to inhibit gliogenic differentiation of neural stem cells and induce neurogenic cell fate commitment [1]. As peak neurogenesis occurs around E14-E17 and gliogenesis is subsequently initiated around E17 [2], the up-regulation of these gliogenesis-suppressing miRNAs around E14 fits well with existing reports. Furthermore, we observed an up-regulation of the gliogenic miR-338 [3] specifically at P0 but not in neurons compared to NPCs, thus underscoring the potential role of this miRNA specifically in gliogenesis. As another example, miR-153 whose inhibition was shown to confer gliogenic competence to neural stem cells [4], was up-regulated in neurons compared to NPCs in our analysis. While our study lacks single cell resolution, these examples demonstrate that we provide a useful reference set for further disentangling the more specific roles of miRNAs in different cell types of the developing brain in future research. We have integrated these points in the revised version of the manuscript (lines 149-160)

4. A schematic diagram summarizing the study may be helpful.

Reply: According to the reviewer's suggestion, we have created a more comprehensive diagram summarizing our study. The schematic is included as Figure 1 in the revised manuscript.

5. Please check again for misspelled words with a spell checker. For example, line 534 says "miNRA differential expression analysis".

Reply: We thank the reviewer for the attention to details. We have performed a spell check in the revised manuscript.

Reviewer #2 (Remarks to the Author):

In this manuscript, the authors detected the dynamic changes of miRNA expression at different time points of brain development by small RNA sequencing, analyzed the co-expression characteristics of miRNAs in the process, and then constructed a co-targeting network by combining with published RNA-seq data, preferably by verifying the targeting of the same gene by in vitro luciferase assays whether there is a synergistic enhancement effect of multiple miRNAs of the same gene. The study looks interesting, but there are

obvious shortcomings in the research strategy that limit the innovativeness of the study, and almost all of the conclusions have been reported similarly before.

Reply: We are grateful to the reviewer for their interest in our study. We believe that the particular strength of our work lies in the high-throughput characterization of expression patterns of miRNAs during cortical development. This makes our study particularly suitable as a comprehensive resource for subsequent, more targeted research. As such, we would argue that replicating several previous findings is an important strength of our analysis. We have provided detailed responses to the questions raised and we hope we could alleviate the reviewer's concerns.

Major points:

1. There is no doubt that the same miRNAs can regulate many target genes, but they selectively act on some of them in different cells or at different stages, with spatiotemporal specificity. The present study is mainly based on sequencing results from E14, E17 and P0 cortices, as well as isolated NPCs from E14.5 cortices and neurons differentiated from these NPCs. Strictly speaking, this study did not effectively distinguish the expression characteristics of miRNAs in different stages and cell types. It is difficult to draw clear and convincing conclusions based on a bulk data analysis of co-expression characteristics and co-targeting networks, especially when E17 and P0 have more diverse cell types and their proportions have also changed.

Reply: The reviewer is correct in pointing out that we cannot draw conclusions about different cell types as we performed bulk sequencing. Thus, we were very careful not to claim that we detect cell type-specific expression patterns of miRNAs and already acknowledged that our samples likely reflect heterogeneous cell populations (lines 145-149). Consequently, we only made inferences about distinct developmental stages (E14, E17 and P0). However, these developmental time points can be taken as proxies for cell type diversity in the developing cerebral cortex. While E14 is a more progenitor-dominated stage, E17 and P0 are indeed associated with an increasing cell type diversity [2]. As we reported in the manuscript (lines 128-129), we observed the biggest transcriptional shift between E14 and E17 which roughly captures the peak neurogenesis phase [2]. However, distinct miRNAs were also differentially expressed in the E17 vs P0 comparison which might correspond to changing expression patterns related to gliogenesis. Though our analysis lacks single cell resolution, as pointed out by the reviewer, we replicated several previous reports on cell-type specific miRNA expression including up-regulation of the neuron-specific miR-124 and miR-137 in neurons and at P0 as well as up-regulation of the intermediate progenitor enriched miR-92a at E14 [5]. It is worthy to mention that several of the developmental-stage dependent expression changes we observed also fit well with previously reported roles of miRNAs in neuro- and gliogenesis, respectively. For instance, miR-106a-5p and its paralog miR-17 were up-regulated at E14 in our analysis and expression levels were significantly reduced at E17/P0. Both miRNAs were previously reported to inhibit gliogenic differentiation of neural stem cells and induce neurogenic cell fate commitment which supports the up-regulation during peak neurogenesis in our study [1]. Furthermore, we observed an up-regulation of the gliogenic miR-338 [3] specifically at P0 but not in neurons compared to NPCs, thus

underscoring the potential role of this miRNA specifically in non-neuronal cells. As another example, miR-153 whose inhibition was shown to confer gliogenic competence to neural stem cells [4], was up-regulated in neurons compared to NPCs in our analysis. While single cell profiling would have ultimately been more informative, this approach remains challenging and bulk sequencing methods are still the standard for high-throughput miRNA profiling. In fact, the only study we are aware of employing a single cell technology is the work by Nowakowski et al. [5]. Since the method employed by Nowakowski and colleagues, however, relies on qPCR, only a pre-selected set of miRNAs can be analyzed. In our study, we aimed at performing a more comprehensive and explorative analysis to identify key miRNAs during murine cerebral cortex development. We believe we therefore provide a valuable resource for more targeted investigations even on the single cell level in subsequent studies.

2. The authors used other published data to analyze mRNA-miRNA and miRNA co-targeting networks, but the time points don't match exactly, and the sample collection may not be completely consistent in different experiments, so why didn't the authors perform transcriptome sequencing under the same conditions when they performed RNA sequencing and then analyze the data in combination with other publicly available data?

Reply: We used the small RNA seq data we generated at E14, E17 and P0 to construct the miRNA-co-expression networks. Co-targeting network prediction is initially independent of miRNA and target gene expression patterns and previous studies have even intentionally not considered expression of miRNAs to generate more general co-targeting predictions [6]. Integrating expression information of miRNAs and target genes was our attempt to put our *in-silico* predictions in a biologically-relevant context. The public developmental RNA seq data we integrated [7] were indeed generated at slightly different time points (E14.5, E16.5 and P0 vs E14, E17 and P0 in our study). However, since in our analysis of integrating validated miRNA targets (Fig. 4e, f) we focus only on differentially expressed genes between E14.5 and P0, we believe that the impact of the time point discrepancy is negligible. As shown on the heatmap below (Reviewer Fig. 1), almost all down-regulated validated targets of green module miRNAs that we included, had a uniformly decreasing expression from E14.5 to P0, thus performing an experiment at E14 instead of E14.5 will likely not change our results appreciably.

Furthermore, it is technically challenging to have completely uniform sampling time points between all individual embryos and performing a new sequencing experiment with the same planned time points as the miRNA sequencing would still be associated with certain sample-to-sample variability as we use embryos from different mothers to have independent biological replicates. This, the exact time of conception might vary slightly. Such variability can also be seen as an advantage as it demonstrates that our results generate well to an independent data set and the biological findings are stronger than technical biases. Ultimately, we are convinced that performing a new RNA sequencing would not change our main findings. Integrating publicly available data sets furthermore aligns well with current research trends relating to the 3R principle in animal research. We hope the reviewer will agree with this approach.

3. It would be more convincing if one of the conclusions of this study were validated *in vivo*, especially for the enhanced gene silencing effect of cooperative binding of co-expressed miRNAs

Reply: Our study primarily focuses on a high-throughput analysis of miRNA longitudinal expression changes during mouse cerebral cortical development. In this respect, we offer several layers of validation of the reported results including the independent sequencing in NPCs and neurons (Supplementary Fig. S2) as well as qPCR measurements in both cerebral cortex samples and NPCs vs. neurons (Supplementary Figure S4). In these experiments, we could replicate our initial results thus confirming their biological validity. Furthermore, luciferase assays are a state-of-the-art approach to investigate the gene silencing effect of miRNAs employed by most studies on this topic. We agree with the reviewer that *in vivo* validation of the cooperative binding would be more convincing. However, such experiments including, e.g., knock-down or overexpression of combinations of miRNAs might be difficult to interpret due to potential off-target effects that would otherwise not be observable under physiological conditions [8]. The presence of binding sites for additional miRNAs in the 3' UTRs of target genes makes co-targeting validation *in vivo* even more difficult.

We would however like to draw the attention of the reviewer to the fact that we independently replicated the cooperative effect of miR-137-3p and miR-153-3p on silencing *Neurod1* expression as reported previously [6]. Additionally, an important prerequisite for miRNA binding and co-regulation is the co-expression of miRNAs and target genes [5] thus our analyses indicate that the co-targeting relationships we observed are in principle possible *in vivo*. We hope this approach is acceptable to the reviewer especially as our study is mostly intended as a resource article.

Reviewer Fig 1. Expression patterns of validated targets of green module miRNAs that were significantly down-regulated at P0 compared to E14.5. Values are shown as z-scores.

Minor points :

1. It is suggested to describe the information such as the sample collection at each time point.

Reply: A detailed description of sample collection and processing is included in the methods section, and we have updated the manuscript according to the reviewer's suggestion (lines 535-552).

2. Figure 6, When verifying the effects of miR-137-3p and miR-153-3p on target genes by luciferase assay, why is the luciferase activity of these two miRNAs alone increased instead?

Reply: This is indeed an interesting observation, however, the exploration of this phenomenon was outside the scope of our study. Possibly, this increase in luciferase activity could be caused by an indirect effect, e.g. the miRNAs regulate other targets that either influence promoter activity of the luciferase constructs or bind to the 3'UTR cloned into the luciferase vector. Alternatively, the interaction might be direct, and the single miRNAs might then possibly activate target expression instead of inhibiting it which is an effect previously reported for some miRNAs [9, 10].

3. The authors repeatedly mention that this is the FIRST comprehensive resource of miRNA longitudinal expression changes during corticogenesis (line 41 and line 414), but there are similar reports, such as reference #48 in the manuscript (Nat Neurosci 21, 1784-1792).

Reply: We apologize for any potential misunderstandings. We wanted to highlight that we provide the first high-throughput resource of miRNA longitudinal expression changes during cerebral cortical development in the mouse. The study mentioned by the reviewer compared miRNA expression in human embryonic brains at GW 15-16 and GW 19-20. We have revised the text in the new version of the manuscript removing the word "first" in the abstract and specifying that our study provides a resource of miRNA expression changes during murine corticogenesis.

References

1. Naka-Kaneda H, Nakamura S, Igarashi M, Aoi H, Kanki H, Tsuyama J, Tsutsumi S, Aburatani H, Shimazaki T, Okano H: **The miR-17/106–p38 axis is a key regulator of the neurogenic-to-gliogenic transition in developing neural stem/progenitor cells.** *Proceedings of the National Academy of Sciences* 2014, **111**:1604-1609.
2. Mukhtar T, Taylor V: **Untangling Cortical Complexity During Development.** *Journal of Experimental Neuroscience* 2018, **12**:1179069518759332.
3. Rajman M, Schratt G: **MicroRNAs in neural development: from master regulators to fine-tuners.** *Development* 2017, **144**:2310-2322.

4. Tsuyama J, Bunt J, Richards LJ, Iwanari H, Mochizuki Y, Hamakubo T, Shimazaki T, Okano H: **MicroRNA-153 Regulates the Acquisition of Gliogenic Competence by Neural Stem Cells.** *Stem Cell Reports* 2015, **5**:365-377.
5. Nowakowski TJ, Rani N, Golkaram M, Zhou HR, Alvarado B, Huch K, West JA, Leyrat A, Pollen AA, Kriegstein AR, et al: **Regulation of cell-type-specific transcriptomes by microRNA networks during human brain development.** *Nature Neuroscience* 2018, **21**:1784-1792.
6. Cherone JM, Jorgji V, Burge CB: **Cotargeting among microRNAs in the brain.** *Genome Res* 2019, **29**:1791-1804.
7. Weyn-Vanhentenryck SM, Feng H, Ustianenko D, Duffié R, Yan Q, Jacko M, Martinez JC, Goodwin M, Zhang X, Hengst U, et al: **Precise temporal regulation of alternative splicing during neural development.** *Nature Communications* 2018, **9**:2189.
8. Kuhn DE, Martin MM, Feldman DS, Terry AV, Jr., Nuovo GJ, Elton TS: **Experimental validation of miRNA targets.** *Methods* 2008, **44**:47-54.
9. Bukhari SIA, Vasudevan S: **FXR1a-associated microRNP: A driver of specialized non-canonical translation in quiescent conditions.** *RNA Biology* 2017, **14**:137-145.
10. Lee S, Vasudevan S: **Post-transcriptional stimulation of gene expression by microRNAs.** *Adv Exp Med Biol* 2013, **768**:97-126.

Stage-specific expression patterns and co-targeting relationships among miRNAs in the developing mouse cerebral cortex

Corresponding Author: Dr Hristo Todorov

Version 0:

Reviewer comments:

Reviewer #1

(Remarks to the Author)

I would like to comment as an experimental researcher. This is an interesting study that provides a comprehensive analysis of the dynamics of miRNAs and their target mRNA networks altered during the progression of neural development. By re-analyzing previous reports and data, this study not only provides logical evidence for the cooperative function of multiple miRNAs, which had been expected from the results of individual miRNA studies, but also provides new findings and is a valuable resource for further study. The manuscript is written in an easy-to-understand manner, even for experimental researchers, and is considered sufficient for publication. Therefore, the following are suggestions, not requirements.

Minor points:

1. It would be helpful to clarify the type of analysis and cell source for each experiment, such as single cell or bulk RNA-seq, cultured cells or tissue-derived non-cultured cells, etc.
2. It might be worth to mention what miRNAs are classified in modules other than black and green, such as yellow, red, and etc.
3. In addition to NPCs vs. neurons, neurogenic NPCs vs. gliogenic NPCs are also known to be important microRNA-regulated transition in neural development. In the case of bulk RNA-seq analysis, the results would show cell heterogeneity-dependent changes in miRNA expression profiles. If you performed the analysis by scRNA-seq data, you could also add the results of changes in miRNA expression profiles associated with changes in neurogenic-to-gliogenic NPCs. In this neurogenic-to-gliogenic transition of NPCs, several reports have shown that the properties of NPCs alter between E14 and E15.
4. A schematic diagram summarizing the study may be helpful.
5. Please check again for misspelled words with a spell checker. For example, line 534 says "miNRA differential expression analysis".

Reviewer #2

(Remarks to the Author)

In this manuscript, the authors detected the dynamic changes of miRNA expression at different time points of brain development by small RNA sequencing, analyzed the coexpression characteristics of miRNAs in the process, and then constructed a co-targeting network by combining with published RNA-seq data, preferably by verifying the targeting of the same gene by in vitro luciferase assays. whether there is a synergistic enhancement effect of multiple miRNAs of the same gene. The study looks interesting, but there are obvious shortcomings in the research strategy that limit the innovativeness of the study, and almost all of the conclusions have been reported similarly before.

Major point:

1. There is no doubt that the same miRNAs can regulate many target genes, but they selectively act on some of them in different cells or at different stages, with spatiotemporal specificity. The present study is mainly based on sequencing results from E14, E17 and P0 cortices, as well as isolated NPCs from E14.5 cortices and neurons differentiated from these NPCs. Strictly speaking, this study did not effectively distinguish the expression characteristics of miRNAs in different stages and cell types. It is difficult to draw clear and convincing conclusions based on a bulk data analysis of co-expression characteristics and co-targeting networks, especially when E17 and P0 have more diverse cell types and their proportions have also changed.

2. The authors used other published data to analyze mRNA-miRNA and miRNA co-targeting networks, but the time points don't match exactly, and the sample collection may not be completely consistent in different experiments, so why didn't the authors perform transcriptome sequencing under the same conditions when they performed RNA sequencing and then analyze the data in combination with other publicly available data?

3. It would be more convincing if one of the conclusions of this study were validated in vivo, especially for the enhanced gene silencing effect of cooperative binding of co-expressed miRNAs

Minor points :

1. It is suggested to describe the information such as the sample collection at each time point.

2. Figure 6, When verifying the effects of miR-137-3p and miR-153-3p on target genes by luciferase assay, why is the luciferase activity of these two miRNAs alone increased instead?

3. The authors repeatedly mention that this is the FIRST comprehensive resource of miRNA longitudinal expression changes during corticogenesis (line 41 and line 414), but there are similar reports, such as reference #48 in the manuscript (Nat Neurosci 21, 1784-1792).

Version 1:

Reviewer comments:

Reviewer #2

(Remarks to the Author)

This revised manuscript has addressed most of my concerns and I am satisfied with the revised version and recommend the manuscript for publication.
